# mHealth Interventions to Address Physical Activity and Sedentary Behavior in Cancer Survivors: A Systematic Review

**DOI:** 10.3390/ijerph18115798

**Published:** 2021-05-28

**Authors:** Selina Khoo, Najihah Mohbin, Payam Ansari, Mahfoodha Al-Kitani, Andre Matthias Müller

**Affiliations:** 1Centre for Sport and Exercise Sciences, University of Malaya, Kuala Lumpur 50603, Malaysia; najihahmohbin@um.edu.my; 2DCU Business School, Dublin City University, Dublin, Ireland; payamansary@gmail.com; 3Physical Education and Sports Sciences Department, College of Education, Sultan Qaboos University, Muscat 123, Oman; mkitani@squ.edu.om; 4Saw Swee Hock School of Public Health, National University of Singapore, Singapore 117549, Singapore

**Keywords:** fitness tracker, exercise, mobile health, mobile application, health behavior

## Abstract

This review aimed to identify, evaluate, and synthesize the scientific literature on mobile health (mHealth) interventions to promote physical activity (PA) or reduce sedentary behavior (SB) in cancer survivors. We searched six databases from 2000 to 13 April 2020 for controlled and non-controlled trials published in any language. We conducted best evidence syntheses on controlled trials to assess the strength of the evidence. All 31 interventions included in this review measured PA outcomes, with 10 of them also evaluating SB outcomes. Most study participants were adults/older adults with various cancer types. The majority (*n* = 25) of studies implemented multicomponent interventions, with activity trackers being the most commonly used mHealth technology. There is strong evidence for mHealth interventions, including personal contact components, in increasing moderate-to-vigorous intensity PA among cancer survivors. However, there is inconclusive evidence to support mHealth interventions in increasing total activity and step counts. There is inconclusive evidence on SB potentially due to the limited number of studies. mHealth interventions that include personal contact components are likely more effective in increasing PA than mHealth interventions without such components. Future research should address social factors in mHealth interventions for PA and SB in cancer survivors.

## 1. Introduction

In 2020, it was estimated that there were about 19.3 million new cancer cases and almost 10.0 million cancer deaths worldwide [1]. Cancer affects all regions of the world and is a major cause of morbidity and mortality. This is concerning considering that it is predicted that an estimated 28.4 million new cancer cases are expected to occur worldwide in 2040. This is a 47% increase in annual cases from the 19.3 million cases reported in 2020 [1].

Although there is a trend of increasing cancer survival [2], cancer survivors must cope with cancer complications and treatments that impact health and quality of life [3]. For example, 25% and 10% of cancer survivors reported poor physical and mental health, respectively. This stands in contrast to 10% and 6% in adults without cancer [4]. Cancer survivors are also at increased risk of recurrent cancer and other diseases, such as cardiovascular disease, diabetes, and osteoporosis [5]. Increasing physical activity (PA) and reducing sedentary behavior (SB) can improve the health and quality of life of cancer survivors.

The value of PA, defined as any bodily movement produced by skeletal muscles that need energy expenditure [6], on cancer survivors have been reported by numerous studies [7]. Engaging in PA after a cancer diagnosis has been found to be associated with decreased risk of cancer-specific and all-cause mortality among individuals with breast, colon, and prostate cancer [8]. Engaging in approximately 150 min per week of moderate PA associated with improved survival rate in breast cancer [9] and colon cancer [10]. In addition, there is strong evidence that engaging in PA during and after cancer treatments is associated with improved physiologic and psychosocial outcomes, including aerobic fitness, body composition, fatigue, mood, and quality of life [11]. This is particularly true for breast and colon cancers.

In addition to studies on PA, SB has received recent research attention. SB, as distinct from insufficient PA, is defined as any waking behavior in a sitting or reclining posture that requires energy expenditure ≤1.5 times the basal metabolic rate [12]. This includes, among others, using electronic devices while sitting, reclining or lying down; sitting at work; reading while lying down, and sitting during transportation. Although limited studies are examining SB and its association with health and wellbeing in cancer survivors, there is evidence that high levels of SB are linked to increased all-cause mortality, sleep disturbance, and depression symptoms in breast cancer survivors and mortality in colorectal cancer survivors [13,14,15].

The American Cancer Society Guidelines on Nutrition and Physical Activity for Cancer Survivors recommend that cancer survivors should engage in at least 150 min of moderate-intensity PA or 75 min of vigorous-intensity PA per week [16]. Unfortunately, many cancer survivors do not meet these guidelines. For example, only 33% of cancer survivors in the United States and 26% in South Korea met the PA recommendation [16,17]. In addition, the World Health Organization recommends to limit SB and/or replace sedentary time with any PA while aiming to do more moderate-to-vigorous PA (MVPA), defined as PA that is performed at >3 total metabolic equivalents (METs), for minimizing detrimental effects of high levels of SB [17]. Many cancer survivors spent a large part of their day being sedentary. Cancer survivors, independent of cancer type, have been found to spend longer than eight hours per day being sedentary [18].

The proliferation of mobile devices has contributed to the popularity of mobile health (mHealth) as a novel mode of delivering health and healthcare [19]. mHealth is defined as using mobile phones, patient monitoring devices, personal digital assistants, and other wireless technologies to support medical and public health practice [20]. The increased ubiquity of mHealth approaches is also apparent in the PA and SB literature, where the number of scientific publications has risen sharply in recent years [21]. Despite this, there are still gaps in behavioral mHealth research. For example, the effects of specific mHealth intervention modalities and components and how mHealth intervention effects may vary by population group are still debatable topics [22]. mHealth interventions have been reported to have small to moderate effects on PA and SB in healthy populations [23].

The evidence on mHealth interventions to address PA and SB among cancer survivors is elusive. To our knowledge, two previous reviews have focused on digital interventions among cancer survivors, and both reported some positive effects only on PA [24,25]. Further, one review focused exclusively on activity tracker interventions among cancer survivors and PA and also showed similar promising effects [26]. None of these reviews have comprehensively analyzed the effects of mHealth interventions on PA and SB in cancer survivors. This is an important gap as extrapolating findings from other populations and intervention modes is not warranted. In terms of populations, this is so because interventions that target cancer survivors need to be specifically designed to account for the distinct challenges to health behavior change they face. These challenges include cancer-related fatigue and lack of support from healthcare providers [27].

The lack of a comprehensive overview of mHealth interventions on PA and SB in cancer survivors warrants the current work. This systematic review aims to identify, evaluate, and synthesize the scientific literature on mHealth interventions to promote PA or reduce SB in cancer survivors. In the Methods section, we outline our inclusion criteria, search strategy, screening procedures as well as analysis approach. The results introduce the descriptive information of the included studies, as well as the data synthesis. Results are discussed in the Discussion section.

## 2. Materials and Methods

This systematic review is registered with the prospective international register of systematic reviews PROSPERO network (registration no. CRD42020167694) and followed the preferred reporting items for systematic reviews and meta-analyses (PRISMA) statement for reporting systematic reviews [28].

### 2.1. Study Inclusion Criteria

#### 2.1.1. Study Designs

Eligible studies employed experimental or quasi-experimental designs, including randomized controlled trials (RCT), controlled trials, pre-and-post trials, and crossover trials.

#### 2.1.2. Participants

Studies were eligible for inclusion if participants were cancer survivors of any age (persons who have been diagnosed with cancer, from the time of diagnosis through the remainder of their lives) [29]. Thus, persons of any age with any type of cancer, either living with cancer, undergoing, or those having completed cancer treatment, were included.

#### 2.1.3. Interventions

This review included studies that implemented an intervention that featured an mHealth component to address PA and/or SB in cancer survivors. mHealth components could have been delivered via mobile devices, such as mobile phones, smartphones, tablets, personal digital assistants, mobile apps, short messaging services, or wearable activity trackers. In this review, we did not consider pedometers as an mHealth component due to their non-interactive nature to communicate electronically with mobile devices or the Internet [30].

#### 2.1.4. Comparator(s)

Studies that compared mHealth interventions with any type of control were included in this review. This may be an electronic health (eHealth) intervention (e.g., a web-based app, email, website), a non-eHealth intervention (e.g., face-to-face, pamphlets, brochures), or a no-intervention control. Studies without a comparison group (e.g., a non-controlled study) were also included.

#### 2.1.5. Outcomes

This review included studies that reported changes in terms of PA or SB following mHealth interventions. For PA, this included changes in energy expenditure, step counts, PA level, daily time PA in minutes/hours, PA frequency, and PA intensity. For SB, this included sitting time/day, sedentary breaks, bouts of prolonged sitting, and screen time. Studies that measured PA and SB outcomes objectively (e.g., via accelerometers) or via self-report (e.g., questionnaires) were included.

### 2.2. Search Strategy

The electronic search strategy aimed to locate published scientific studies in any language from 2000 until 13 April 2020. The year 2000 was chosen because a previous bibliometric analysis of studies on e- and mHealth for PA, SB, and diet showed that almost no studies were published before that year [21]. The following six databases were systematically searched: Web of Science, PubMed, Scopus, CINAHL, Cochrane Central Register of Controlled Trials (CENTRAL), and SPORTDiscus. The search strategy presented in Appendix A consists of three main categories, namely (1) “cancer” population, (2) “mHealth” intervention method, and (3) “physical activity” or “sedentary behavior” outcome variable and their synonym keywords in each category. Reference lists of relevant articles were also searched to identify additional articles. Unpublished studies, preprints and gray literature were not included in this review.

### 2.3. Study Selection

Following the search, references were imported into Zotero 5.0.89 (Corporation for Digital Scholarship, Vienna, Virginia 22182, USA), and duplicates were removed. Two independent reviewers (N.M., M.A.) screened titles and abstracts of potentially relevant articles against the inclusion criteria for the review. Disagreements were resolved by a third reviewer (A.M.M.) with vast experience in the research area and systematic review methods. Full texts of potentially relevant papers were retrieved and screened similarly. As only articles published in English were included in full-text screening, no interpretation was conducted for different language articles. The number of articles at each screening stage is shown in Figure 1.

### 2.4. Data Extraction

Data were extracted from included studies by one reviewer (N.M.) and checked for completeness and accuracy by a second reviewer (P.A.) using a standardized data extraction form adapted from a checklist presented in the *Cochrane Handbook for Systematic Reviews of Interventions* [31]. The extracted information of included studies included: author, year of publication, country of study, study aim, study design, sample characteristics, intervention characteristics, comparator information, outcomes measured, and results. Any disagreements that arose were resolved through consultation with a third reviewer (A.M.M.).

### 2.5. Risk of Bias Appraisal

Included studies were critically appraised by two independent reviewers (S.K., N.M.) using the revised Cochrane risk-of-bias tool for randomized trials (RoB 2) [32] and the risk of bias in nonrandomized studies of interventions (ROBINS-I) [33]. RoB 2 includes the following domains: randomization process, deviations from intended interventions, missing outcome data, measurement of the outcome, and selection of reported results. ROBINS-I contains seven domains, including bias due to confounding, bias in the selection of participants into the study, bias in classification of interventions, bias due to deviations from intended interventions, bias due to missing data, bias in the measurement of outcomes, and bias in the selection of the reported result. Any disagreements that arose between the reviewers were resolved through discussion with a third reviewer (A.M.M.).

### 2.6. Analysis

Study and intervention characteristics are described narratively. We decided against conducting a meta-analysis due to the significant heterogeneity of eligible studies in terms of intervention length, design, comparators, and outcomes. We categorized studies based on study design: controlled and non-controlled trials. We further grouped them according to intervention components: interventions that only used an mHealth component; interventions that used mHealth in addition to non-mHealth components that involved no personal contact (e.g., printed material or a website); interventions that used mHealth in addition to non-mHealth components that involved personal contact (e.g., group meetings, phone consultations).

To establish the effectiveness of interventions based on the above categorization, we conducted best evidence synthesis [34] following guidelines by Fanchini et al. [35] and Kuijer et al. [36], which were adapted from van Tulder et al. [37]. We only considered controlled trials when evaluating the evidence due to their inherently lower risk of bias. We considered the risk of bias of these controlled studies when judging the evidence. For this, the proportion of RCT with low risk of bias and some concerns raised as well as quasi-experimental trials with low risk of bias, were the basis for judging the strength of the evidence. We decided that low-risk of bias quasi-experimental trials are likely comparable to RCT with some risk [33]. We defined the following adapted evidence categories and accompanying criteria from previous guidelines [35,36,37].
(1)Strong evidence: at least 66% (2/3) of controlled trials with low/some risk of bias show effect in the same direction;(2)Moderate evidence: 50% to 65% of controlled trials with low/some risk of bias show effect in the same direction;(3)Limited evidence: less than three low/some concerns risk of bias controlled trials available;(4)Inconclusive evidence: other findings not applicable to strong, moderate, or limited. For example, inconsistent findings in multiple studies (two of five or 40% low-risk studies show effect in the same direction).

## 3. Results

### 3.1. Study Selection

A total of 5971 articles were identified, and after the removal of duplicates, 3917 articles were screened for eligibility, of which 3839 were excluded. Following the first screening of titles and abstracts, 78 articles were eligible for full-text screening. After the full-text screening, 32 articles reporting on 31 interventions were included in this review. The study by Lynch et al. [38] was reported in another article [39]. Thus both will be reported as one study in this review. The PRISMA flowchart (see Figure 1) summarizes the study selection.

### 3.2. Study Characteristics

Of the 31 included studies, 14 were conducted in the United States [10,40,41,42,43,44,45,46,47,48,49,50,51,52], five in Australia [38,53,54,55,56], four in Republic of Korea [57,58,59,60], two each in Canada [61,62] and France [63,64], and one each in The Netherlands [65], Germany [66], Spain [67], and United Kingdom [68]. All studies were conducted between 2015 and 2020. Sixteen of the included studies were RCT [38,40,42,44,46,47,48,49,51,52,53,56,61,64,65,69], whereas 12 studies employed a pre-post [41,43,45,50,54,55,59,60,62,63,67,68], and three studies applied a quasi-experimental design [57,58,66]. The study duration ranged from two weeks [55] to six months [41,45,46,48,51,54,63]. Comparator groups included active controls in 10 studies [42,46,49,51,53,56,57,64,66,69] and inactive controls in eight studies [38,40,44,47,52,58,61,65]. In one study where more than one comparator group was used, one control group was active and the other inactive [48].

### 3.3. Participant Characteristics

A total of 1977 participants were recruited into the included studies at baseline, with a sample size ranging from 10 [55] to 356 [57]. Most study participants were adults/older adults with two studies, including only children/adolescents [40,45] and another study, including adolescents/young adults [41]. About a third of the studies were conducted among breast cancer survivors [38,49,51,52,53,54,57,58,61,63,67], two studies mainly included prostate cancer survivors [47,62], and another two studies included colorectal/colon cancer survivors [46,59,69]. Eight studies included survivors of various cancer types [40,41,43,44,55,64,65,68]. Several studies focused on two cancer types, such as colorectal cancer with gynecologic cancer survivors [56], breast cancer with colorectal cancer survivors [42] and breast cancer with endometrial cancer survivors [50]. The remaining studies were conducted among survivors of brain tumors [45], endometrial cancer [48], pediatric cancer [66], and hepatocellular carcinoma [60].

### 3.4. Intervention Characteristics

The majority (25/31) of the included studies implemented multicomponent interventions where the mHealth component was combined with or without personal contact [38,40,41,42,43,44,45,46,47,48,49,50,51,52,53,54,55,56,57,61,62,63,64,66,69]. Six studies exclusively relied on mHealth intervention components [58,59,60,65,67,68]. The most commonly used mHealth technology was activity trackers with 10 studies using only activity trackers [41,42,45,49,52,53,55,56,62,66], six studies using activity trackers and affiliated app [38,51,59,60,61,63], and four studies combined activity trackers with text messages [43,44,47,69]. Eight studies featured a smartphone app in their interventions [46,50,55,57,58,65,67,68], whereas three studies used text messages [48,54,64]. One study combined multiple mHealth components, namely activity trackers, app, and text messages [40]. Details of the included studies are presented in Table 1.

### 3.5. Intervention Effects on PA Outcomes

All 31 included studies measured PA outcomes. MVPA was the most frequently reported PA outcome (*n* = 18) [38,40,41,42,43,44,46,47,49,52,53,54,55,56,61,62,67,69], followed by total activity (*n* = 13) [42,48,49,50,51,52,53,57,59,60,61,63,65], and step counts (*n* = 11) [42,43,45,47,49,53,58,62,64,67,69].

#### 3.5.1. Intervention Effects on PA Outcomes in Controlled Trials

Out of eight controlled trials, seven reported significant effects of mHealth interventions with personal contact on MVPA [38,42,44,52,53,56,61]. From these trials, two interventions featuring activity trackers and apps with personal contact reported a significant increase in MVPA daily minutes [38,61] when compared to inactive control groups. Similarly, interventions that combined activity trackers with personal contact showed significant MVPA increases compared to both active [42,53,56] and inactive [52] controls. In an intervention that used activity trackers and text messages in conjunction with personal contact, MVPA improved significantly compared to the inactive control group [44].

Some controlled trials also reported PA outcomes as total PA (*n* = 6) [42,48,52,53,61,65], total energy expenditure [49,51], or METs [57]. From these studies, only two mHealth interventions showed a significant increase in total activity [52,53]. These interventions employed activity trackers and also had a personal contact component. However, there was no effect on total PA when activity trackers were the only intervention component [49].

Step counts were reported as an outcome in seven controlled trials [42,47,49,53,58,64,69]. Only two of these controlled trials reported significant increases in step counts of interventions. One intervention only featured mHealth components [58], and the other intervention had other non-mHealth components without personal contact [42]. Further details are reported in Table 2, where interventions are categorized by intervention components: mHealth only interventions, multicomponent interventions featuring mHealth and other non-personal contact components, multicomponent interventions featuring mHealth and personal contact components.

#### 3.5.2. Effects on PA Outcomes in Non-Controlled Trials

In total, six non-controlled trials reported on MVPA, of which two reported a significant effect favoring the mHealth intervention [62,67]. A mHealth only intervention featuring an app to monitor and provide feedback on PA increased MVPA [67]. Another intervention that used activity trackers in conjunction with other non-personal contact intervention components showed significant MVPA improvements [62].

Total PA [50,63] and the total metabolic equivalent [59,60] as outcomes were reported in four non-controlled trials. Out of these, only one intervention that relied solely on mHealth components leads to increased total energy expenditure [60]. This finding is in contrast to a similar study, which employed a similar intervention approach and found no effects on energy expenditure [59]. Step counts were also reported in four noncontrolled trials [43,45,62,67]. Only one of these reported significant effects on weekly step counts following an intervention [62]. Further details are reported in Table 3.

### 3.6. Intervention Effects on SB Outcomes

Ten out of 31 of included studies reported on the effects of mHealth interventions on SB outcomes [38,40,41,44,49,56,61,62,63,65]. Overall, only four studies reported a significant decrease in SB following an mHealth intervention. Two controlled trials that used activity trackers with apps and personal contact reported decreased daily sedentary time when compared to inactive comparators [38,61]. Reduction in weekly sedentary time was also reported in a non-controlled trial in which an intervention featuring an activity tracker and web-based components was implemented [62]. Finally, one non-controlled study reported a significant reduction in weekly sitting time following an mHealth intervention [63].

### 3.7. Risk of Bias Assessment

Ten out of 16 RCT were considered to have a low risk of bias. Four RCT were judged to have some concerns arising from the randomization process [46], outcome measurement [53,65], and selection of the reported result [44]. A high risk of bias was present in two RCT, which had several concerns due to lack of information in the randomization process [48,64] and selective in reporting the results [64]. The study by Haggerty et al. [48] had a high risk of bias due to a lack of information on subjective measurement of outcome. Of the non-RCT included, there were three quasi-experimental [57,58,66] and 12 pre-post studies [41,43,45,50,54,55,59,60,62,63,67,68]. All the non-RCT had either high-risk of bias [45,54,59,63,66,68] or no information [41,43,50,55,58,60,62,67] except one in which risk was slightly lower [57]. Details of the risk of bias assessment are presented in Appendix A.

### 3.8. Best Evidence Synthesis

There is strong evidence that mHealth interventions in conjunction with personal contact components can lead to increases in MVPA among cancer survivors based on seven out of eight (87.5%) controlled trials with either low-risk or bias [38,42,52,56,61] or some concerns raised [44,53] reporting positive effects. There is inconclusive evidence on mHealth interventions with personal contact to impact total PA/activity among cancer survivors based on two out of six (33.3%) controlled trials with low risk of bias [52] and some concerns raised [53] reporting positive effects. There is limited evidence on the effectiveness of mHealth interventions with a personal contact for step counts among cancer survivors, as only one out of two controlled trials with low-risk reported an increase in step counts [42]. There is inconclusive evidence that mHealth interventions with personal contact are effective in reducing sedentary time among cancer survivors, as only two out of six (33.3%) low-risk-controlled trials showed significant effects. There is limited evidence on the effects of mHealth only interventions due to a lack of studies. In addition, there is inconclusive evidence on the effects of mHealth interventions without personal contacts components for PA due to inconsistent findings in studies. There were no studies, which examined the effects of mHealth interventions without personal contacts components for SB, and this hindered the evidence synthesis.

## 4. Discussion

The aim of this review was to identify, evaluate, and synthesize the scientific literature on mHealth interventions to promote PA or reduce SB in cancer survivors. In brief, 31 studies of mHealth interventions were included and systematically analyzed. All the studies evaluated the effects of mHealth interventions on PA outcomes, and only 10 of these also reported on SB outcomes. To evaluate the effects of mHealth interventions, we conducted the best evidence synthesis for which we only considered controlled trials due to their inherently lower risk of bias. The synthesis revealed that only for interventions that employed mHealth components in conjunction with personal contact components, there was strong evidence of effects on MVPA.

From our best evidence synthesis, it suggests that the implementation of mHealth interventions with personal contact may be effective in increasing MVPA among cancer survivors. This finding is consistent with a previous meta-analysis that reported digital interventions to increase MVPA by approximately 40 min per week among cancer survivors [25]. It is also in line with reviews on eHealth interventions in cancer survivors [24,70]. However, these earlier reviews did not examine the effects of how interventions were implemented (mHealth only, mHealth plus non-personal contact components, mHealth plus personal contact components), and as such, direct comparisons are difficult to make. This mHealth interventions that also employed personal contact components increased MVPA is intriguing because it suggests that in-person sessions, phone calls, or group consultations are to be considered when designing interventions. It is probable that such an approach will yield greater effects on health behaviors as research has shown a link between social support from others and PA in cancer survivors.

Cancer survivors whose social interactions were limited tend to engage in less PA [71]. Cancer survivors have specific support needs [27] that may not be fulfilled by purely digital interventions. Incorporating personal contact elements into mHealth interventions can be done in various ways. Personal contact in this review was defined as any person-to-person contact involving nondigital/traditional modes of communication (i.e., phone calls and face-to-face meetings). Lynch et al. [38] incorporated an in-person goal-setting session and phone call behavioral counseling to facilitate adaptation to and maintenance of PA. It is likely that these interactions provided social support because they allowed participants to receive feedback and encouragement [72]. Social support through direct interaction has repeatedly been shown to positively influence PA in various populations [73]. Other interventions included follow-up discussions [61], over the phone as well as in-person counseling [53], and goal-setting sessions [42], which all fall within the domain of social support.

Researchers that assessed an mHealth intervention and its effects on MVPA in controlled studies were often 12 weeks long. Most reported significant effects of mHealth interventions that incorporated personal contact after this time and for this outcome [38,42,52,53,56,61]. However, one 8-weeks controlled study with a similar intervention mode also showed increased MVPA [44]. Although it is not possible to provide an intervention duration that is optimal for increasing PA, 12 weeks appears to be a reasonable length that may also be feasible to implement. However, it is important to consider that the intervention duration may be less important than engaging participants effectively [74]. Interventions of short duration may be highly effective if they greatly engage participants.

There was inconclusive or limited evidence regarding the other intervention modes (mHealth only and mHealth without personal contact) and other outcomes. As such, it is not clear whether cancer survivors will benefit from mHealth interventions with or without other components in terms of total activity, step counts, and SB. It was surprising that a very limited number of studies evaluated interventions that solely relied on mHealth components. However, findings from only mHealth interventions in three non-controlled trials showed promising improvement in PA outcomes [60,67,68], warranting more research.

Despite a plethora of research on mHealth interventions targeting PA, there is still limited research on interventions targeting SB. There was inconclusive evidence in relation to mHealth interventions with personal contacts on SB. Controlled studies that incorporated this mode of intervention mainly only target PA behavioral changes and showed no effects on SB [44,49]. In contrast, findings from mHealth interventions with [63] and without [62] personal contacts targeting both PA and SB behavioral changes in non-controlled trials showed a promising reduction in sedentary time. However, SB is distinct from physical inactivity [75], whereas sitting time is unassociated with inadequate amounts of MVPA [76]. Further, it is reported that sitting time takes the most proportion of the waking day and displaces light–intensity PA [77]. Thus, a feasible approach to reduce SB among cancer survivors could be conducted in the future by replacing sitting time with light-intensity PA before gradually shifting to MVPA.

Findings from this review may not be generalizable to all cancer survivors because studies were only conducted in high-income countries. This is disappointing and indicates that the inherent potential of mHealth interventions in many lower-income countries has not yet been utilized. Using the available mobile network infrastructure in many countries to support cancer survivors in resource-constrained contexts should be considered. Although there may still be barriers in terms of the accessibility of more advanced personal mobile devices, such as fitness trackers [78], lower-end technology can also be utilized. This digital behavioral intervention can yield positive effects in middle- to low-income countries, as has been reported previously [79].

This review has several strengths. An extensive search strategy in six large databases using broad search terms was conducted. The procedures of this review were in line with the PRISMA guidelines, which strengthens the methodology of the review. We also included all experimental studies, which assessed PA and/or SB outcomes following the breadth of mHealth interventions in cancer survivors. This leads to a more comprehensive overview of mHealth as an intervention to promote PA and reduce SB for cancer survivors. In this review, the findings were grouped into intervention modes (mHealth only, mHealth plus non-personal contact components, mHealth plus personal contact components) and their effect on the outcomes. Limitations of this review are the variety of study designs, including several with a small sample size due to its pilot nature. This systematic review shows the heterogeneity in the methodology and study designs of the selected studies likely become its weakness. These issues make a comparison between selected studies difficult and why a meta-analysis could not be conducted.

In the future, larger studies with higher quality study designs are needed to generate findings that encompass the whole spectrum of movement behaviors and mHealth interventions. The effects of mHealth interventions targeting SB are still unclear. Thus more studies focusing on mHealth interventions reducing SB should be conducted. As only studies from high-income countries were included in this review, it is uncertain if the findings also apply to middle- to low-income countries. Social support when paired with mHealth intervention components has potential for promoting PA among cancer survivors, while effects on SB are still elusive. It is recommended to investigate the cost-effectiveness of mHealth interventions implementation with virtual or non-virtual social aspects in-depth. Finally, large-scale implementations, which consider a thorough cost-effectiveness analysis of mHealth interventions targeting both PA and SB in all ranges of income countries, require attention in future mHealth research involving cancer survivors.

## 5. Conclusions

We systematically reviewed the scientific literature on mHealth interventions that aimed to promote PA and/or reduce SB in cancer survivors. mHealth interventions with personal contact appeared to have a positive effect on MVPA among cancer survivors. The evidence for this observation was strong. Further, mixed findings for other PA outcomes like total activity and step counts were observed, while the evidence on SB outcomes was inconclusive due to the lack of studies. More research is needed to establish the optimal mHealth intervention mode for various PA outcomes in cancer survivors. Interventions that aim to reduce SB among cancer survivors are also highly encouraged as cancer survivors may be more likely to engage in changing lower-threshold health behavior. Finally, researchers may focus on the cost-effectiveness of interventions because mHealth interventions may need to incorporate personal contact components, which is more costly.

## Figures and Tables

**Figure 1 ijerph-18-05798-f001:**
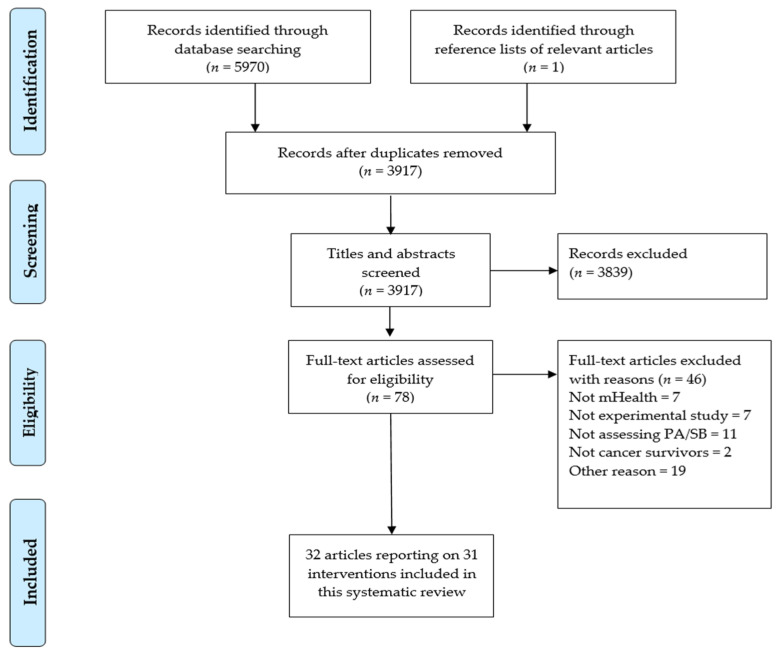
PRISMA diagram illustrating the flow of records.

**Table 1 ijerph-18-05798-t001:** Characteristics of included studies.

First Author, Year, Country	Study Design	Sample Size; Age Group; Cancer Type(s)	Intervention Group	Control Group	PA/SB Outcome	Other Outcome
Cadmus-Bertram, 2019, United States [42]	12 weeks RCT (Pilot)	Baseline: *n* = 50 Analyzed: *n* = 47; Adults/older adults; Breast, colorectal	mHealth: Activity tracker non-mHealth component: Printed material, in-person session, social support, email-based coaching and electronic health record linked to an activity tracker	Printed material and email	PA	Weight
Cheong, 2018, Republic of Korea [59]	12 weeks pre-post	Baseline/analyzed: *n* = 75; Adults/older adults; Colorectal	mHealth: app providing daily exercise program and activity tracker to record activity	Not applicable	PA	Muscle strength, cardiorespiratory fitness, QoL
Chung, 2019, Republic of Korea [58]	12 weeks Quasi-experimental	Baseline: *n* = 54 Analyzed: *n* = 37; Adults/older adults; Breast	mHealth: WalkOn R app to promote healthy activities in a mobile community	No intervention	PA	Not applicable
Delrieu, 2020, France [63]	6 months Pre-Post	Baseline: *n* = 49 Analyzed: *n* = 44; Adults/ older adults; Breast	mHealth: Activity tracker and affiliated app non-mHealth component: In-person or phone sessions on performance feedback or recommendations of PA and SB	Not applicable	PA, SB	Cardiorespiratory fitness, strength, anthropometry, QoL, fatigue
Gell, 2017, United States [43]	4 weeks pre-post (Pilot)	Baseline/Analyzed: *n* = 24; Adults/ older adults; Various	mHealth: Activity tracker and text messages non-mHealth component: Health coaching phone calls	Not applicable	PA	Self-regulation, fatigue, depression, acceptability
Gell, 2020, United States [44]	8 weeks RCT (Pilot)	Baseline: *n* = 66 Analyzed: *n* = 59; Adults/older adults; Various	mHealth: Activity tracker and text messages non-mHealth component: Phone calls from health coach	No intervention	PA, SB	Adherence wearing activity tracker, intervention satisfaction
Götte, 2018, Germany [66]	10 weeks Quasi-experimental	Baseline: *n* = 40 Analyzed: *n* = 39; Children/adolescents; pediatric	mHealth: Activity tracker non-mHealth component: Supervised exercise intervention during and after acute cancer treatment	Exercise intervention after acute cancer treatment	PA	QoL, motor performance, acceptability
Haggerty, 2017, United States [48]	6 months RCT (Pilot)	Baseline: *n* = 41 Analyzed: *n* = 32; Adults/older adults; Endometrial	mHealth: Text messages to support weight loss non-mHealth component: Conventional weighing scale	Telemedicine (active CG) and no intervention (inactive CG)	PA	Weight loss, anthropometry, QoL
Hartman, 2018, United States [52]	12 weeks RCT	Baseline/Analyzed: *n* = 87; Adults/ older adults; Breast	mHealth: Activity tracker non-mHealth component: In- person session to set PA goals	No intervention	PA	BMI, neurocognitive functioning
Kenfield, 2019, United States [47]	12 weeks RCT (Pilot)	Baseline: *n* = 76 Analyzed: *n* = 65; Adults/older adults; Prostate	mHealth: Activity tracker and text messages non-mHealth component: Website providing behavioral and social support information	No intervention	PA	Acceptability
Kim, 2020, Republic of Korea [60]	12 weeks pre-post	Baseline/Analyzed: *n* = 31; Adults/older adults; Hepatocellular carcinoma	mHealth: Care app and activity tracker prescribing exercise program	Not applicable	PA	Cardiorespiratory fitness, strength, anthropometry, QoL
Le, 2017, United States [41]	6 months Pre-post (pilot)	Baseline: *n* = 19 Analyzed: *n* = 15; Adolescents/young adults; Various	mHealth: Activity tracker to record PA non-mHealth component: Website and instruction to adhere to exercise recommendations	Not applicable	PA	Cardiorespiratory fitness, barriers to exercise
Lozano-Lozano, 2019, Spain [67]	8 weeks pre-post	Baseline: *n* = 80 Analyzed: *n* = 76; Adults/older adults; Breast	mHealth: BENECA mHealth app to monitor and provide feedback on healthy eating and PA	Not applicable	PA	QoL, self-efficacy, anthropometry
Lynch, 2019, Australia [38,39]	12 weeks RCT	Baseline: *n* = 83 Analyzed: *n* = 80; Adults/older adults; Breast	mHealth: Activity tracker and affiliated app non-mHealth component: In-person session and telephone-health coaching sessions	No intervention	PA, SB	Not applicable
Maxwell–Smith, 2019, Australia [56]	12 weeks RCT	Baseline: *n* = 68 Analyzed: *n* = 67; Adults/older adults; Colorectal, gynecologic (at risk of cardiovascular disease)	mHealth: Activity tracker non-mHealth component: Dashboard to collect PA engagement data, printed material, group sessions and phone call at Week 8	Printed material	PA, SB	BMI, blood pressure
Mayer, 2018, United States [46]	6 months RCT	Baseline: *n* = 284 Analyzed: *n* = 227; Adults/older adults; Colon	mHealth: SurvivorCHESS app to help increase daily activity levels non-mHealth component: Printed material, self-learning audio program for cancer survival and pedometer	Printed material, self-learning program for cancer survival and pedometer	PA	QoL, distress
McCarroll, 2015, United States [50]	4 weeks pre-post	Baseline: *n* = 50 Analyzed: *n* = 35; Adults/older adults; Breast, endometrial	mHealth: LoseIt! app to support weight-loss non-mHealth component: Weighing scale to track weight	Not applicable	PA	QoL, self-efficacy, anthropometry
McNeil, 2019, Canada [61]	12 weeks RCT (Pilot)	Baseline: *n* = 45 Analyzed: *n* = 41; Adults/older adults; Breast	mHealth: Activity tracker and app to prescribe exercise non-mHealth component: Diary and phone call or e-mail	No intervention	PA, SB	Anthropometry, cardiorespiratory fitness
Mendoza, 2017, United States [40]	10 weeks RCT (Pilot)	Baseline/analyzed: *n* = 59; Children/adolescents; Various	mHealth: Activity tracker with app and text messages non-mHealth component: Facebook support group and phone call to set a daily step goal	No intervention	PA, SB	QoL, acceptability
Ormel, 2018, Netherlands [65]	12 weeks RCT	Baseline/analyzed: *n* = 32; Adults/older adults; Various	mHealth: RunKeeper app for self-monitoring PA	No intervention	PA, SB	Usability of app
Ovans, 2018, United States [45]	12 weeks Pre-post (pilot)	Baseline/analyzed: *n* = 15; Children/ adolescents; Brain tumor	mHealth: Activity tracker non-mHealth component: Phone or in-person coaching to encourage PA	Not applicable	PA	Cardiorespiratory fitness, QoL, fatigue
Pope, 2018, United States [49]	10 weeks RCT	Baseline: *n* = 30 Analyzed: *n* = 20; Adults/older adults; Breast	mHealth: Activity tracker non-mHealth component: Facebook group providing PA tips	Facebook group	PA, SB	Anthropometry, cardiorespiratory fitness
Puszkiewic, 2016, United Kingdom [68]	6 weeks pre-Post	Baseline/Analyzed: *n* = 11; Adults/older adults; Various	mHealth: GAINFitness app that provides a PA program	Not applicable	PA	Anthropometry, QoL, sleep quality, app engagement
Short, 2018, Australia [55]	2 weeks pre-post (pilot)	Baseline/analyzed: *n* = 10; Adults/older adults; Various	mHealth: app to support PA. non-mHealth component: In- person consultation, handout, and telephone or email.	Not applicable	PA	Acceptability
Singh, 2020, Australia [53]	12 weeks RCT	Baseline: *n* = 52 Analyzed: *n* = 50; Adults/older adults; Breast (after completed supervised exercise intervention)	mHealth: Activity tracker non-mHealth component: Counselling session and printed material	Counselling session and printed material	PA	Acceptability
Spark, 2015, Australia [54]	6 months pre-post	Baseline: *n* = 25 Analyzed: *n* = 23; Adults/older adults; Breast	mHealth: Text messages to promote weight loss, PA and dietary behavior change non-mHealth component: Phone call to tailor self-regulation strategies	Not applicable	PA	Weight
Trinh, 2018, Canada [62]	12 weeks pre-post (pilot)	Baseline/Analyzed: *n* = 46, Adults/older adults; Prostate	mHealth: Activity tracker non-mHealth component: Web-based SB intervention	Not applicable	PA, SB	QoL, feasibility
Uhm, 2017, Republic of Korea [57]	12 weeks Quasi-experimental	Baseline: *n* = 356 Analyzed: *n* = 339; Adults/older adults; Breast	mHealth: Smart After Care exercise app non-mHealth component: Pedometer	Printed material	PA	BMI, blood pressure, heart rate, strength, cardiorespiratory fitness, QoL, user satisfaction
Valle, 2017, United States [51]	6 months RCT (Pilot)	Baseline: *n* = 35 Analyzed: *n* = 33; Adults/older adults; Breast	mHealth: Activity tracker interfaced with app non-mHealth component: In-person session, wireless weighing scale, and email-delivered behavioral lessons	Self-regulation intervention group and wireless weighing scale	PA	BMI
Van Blarigan, 2019, United States [69]	12 weeks RCT (Pilot)	Baseline: *n* = 41 Analyzed: *n* = 39; Adults/older adults; Colorectal	mHealth: Activity tracker and daily text messages non-mHealth component: Printed material on PA after cancer	Printed material on PA after cancer	PA	Feasibility and acceptability
Villaron, 2018, France [64]	8 weeks RCT (Pilot)	Baseline/Analyzed: *n* = 43; Adults/older adults; Various	mHealth: Weekly PA encouraging text messages non-mHealth component: PA recommendations, pedometer and printed material	Pedometer	PA	QoL, fatigue

BMI: body mass index; CG: control group; PA: physical activity; QoL: quality of life; RCT: randomized control trial; SB: sedentary behavior.

**Table 2 ijerph-18-05798-t002:** Physical activity (PA) and sedentary behavior (SB) intervention effects were reported in controlled trials.

**mHealth Only Interventions**							
**First Author, Year**	**mHealth Intervention**		**Control**	**Effects**			**Risk of Bias**
				**PA**	**SB**	**Results Summary**	
Chung, 2019 [58]	App to motivate and provide information on PA, healthy diet and distress		Inactive	*Significant effect* Step counts (steps/w) ^O^	Not applicable	Significant between-group increase in step count favoring the intervention group	No information ** (NI,?,+,+,?,?,+)
Ormel, 2018 [65]	App for PA self-monitoring		Inactive	*No significant effect* Total PA (min/w) ^S^, Physical activity scale for the elderly (sum score) ^S^	*No significant effect* Sitting time (min/w) ^S^	No significant between-group change in either outcome	Some concerns * (+,+,+,?,+)
**Multicomponent interventions with mHealth component**							
**First Author, Year**	**Intervention**		**Control**	**Effects**			**Risk of Bias**
	**mHealth**	**Non-mHealth Component**		**PA**	**SB**	**Results Summary**	
**Multicomponent intervention without personal contact (e.g., No phone and in-person contact)**							
Haggerty, 2017 [48]	Daily text messages providing feedback, support, and strategies to adhere to behavior change	Conventional weighing scale	Active and inactive	*Significant effect* Walking activity, vigorous PA (METs/w) ^S^ *No significant effect* Total PA (METs/w) ^S^	Not applicable	Significant between-group increase in walking favoring intervention group with a significant increase of vigorous PA in favor of the inactive control group	High * (+,+,?,?,+)
Kenfield, 2019 [47]	Activity tracker and text messages to motivate behavioral changes following recommendations	Website	Inactive	*No significant effect* MVPA (min/d) ^O^, Step counts (steps/d) ^O^	Not applicable	No significant between-group changes in either outcome	Low * (+,+,+,+,+)
Mayer, 2018 [46]	Apps providing information and support to increase daily activity levels	Printed material, self-learning audio program and pedometer	Active	*No significant effect* MVPA (min/w) ^S^	Not applicable	No significant between-group changes in outcome	Some concerns * (?,+,+,+,+)
Pope, 2018 [49]	Activity tracker monitoring PA and health metrics	Facebook group	Active	*No significant effect* Light PA, MVPA (min/d) ^O^, Energy expenditure (kcal/d) ^O^, Step counts (Steps/d) ^O^	*No significant effect* SB (min/d) ^O^	No significant between-group changes in either outcome	Low * (+,+,+,+,+)
Uhm, 2017 [57]	Apps providing information and monitor the prescribed exercises	Pedometer and printed material	Active	*No significant effect* Total metabolic equivalent (METs/w) ^S^	Not applicable	No significant between-group changes in outcome	Serious ** (/,?,+,+,+,?,+)
Van Blarigan, 2019 [69]	Activity tracker to assess PA and daily text messages providing PA information	Printed material	Active	*No significant effect* MVPA, moderate PA, vigorous PA (min/d) ^O^, Step counts (steps/d) ^O^	Not applicable	No significant between-group changes in either outcome	Low * (+,+,+,+,+)
Villaron, 2018 [64]	Text messages on recommendations to increase PA	Printed material	Active	*No significant effect* Step counts (steps/w) ^O^	Not applicable	No significant between-group changes in outcome	High * (?,+,+,+,?)
**Multicomponent intervention with personal contact**							
Cadmus-Bertram, 2019 [42]	Activity tracker for self-monitoring PA and developing self-regulatory skills	Printed material, in-person session, social support and email	Active	*Significant effect* MVPA, moderate PA (min/w)^O^, MVPA in bouts (min/d) ^O^, Step counts (steps/d) ^O^ *No significant effect* Total PA, vigorous PA, light PA (min/w) ^O^	Not applicable	Significant between-group increase in MVPA, moderate PA and step counts favoring the intervention group	Low * (+,+,+,+,+)
Gell, 2020 [44]	Activity tracker for self-monitoring PA level and text messages to support PA engagement	Phone call	Inactive	*Significant effect* MVPA (min/w) ^O^ *No significant effect* Light PA (min/d) ^O^, Adjusted light intensity time (% of day) ^O^	*No significant effect* Sedentary time (min/d ^O^) Adjusted sedentary time (% of day ^O^)	Significant between-group increase in MVPA favoring the intervention group	Some concerns * (+,+,+,+,?)
Götte, 2018 [66]	Activity trackers assessing and providing feedback on PA during and after cancer treatment	In-person session	Active	*No significant effect* Daily step goal, active time goal (Goals achievement percentage,?) ^O^	Not applicable	No significant between-group changes in either outcome	High ** (?,?,?,+,+,?,+)
Hartman, 2018 [52]	Activity tracker promoting behavior change	Phone calls and emails	Inactive	*Significant effect* MVPA, total activity (min/d) ^O^, ?Number of participants meeting 150 min/w (n,?)	Not applicable	Significant between-group increase in MVPA, total activity and number of participant meeting 150 min/w favoring the intervention group	Low * (+,+,+,+,+)
Lynch, 2019 [38,39]	Activity tracker with the app providing inactivity alerts and assess PA	In-person session and phone call coaching	Inactive	*Significant effect* MVPA (min/w) ^O^	*Significant effect* Sitting time (min/d) ^O^	Significant between-group increase MVPA and decrease sitting time favoring the intervention group	Low * (+,+,+,+,+)
Maxwell–Smith, 2019 [56]	Activity tracker to record the daily activity and as an encouragement to increase PA target	Printed material, group sessions and phone call	Active	*Significant effect* MVPA, moderate PA (min/w) ^O^ *No significant effect* Proportion of MV10: MVPA accrued in bouts of at least 10 min (min/d) ^O^	*No significant effect* SB (hours/w) ^O^	Significant between-group increase in MVPA and moderate PA favoring the intervention group	Low * (+,+,+,+,+)
Mcneil, 2019 [61]	Activity tracker with app prescribing exercise intensity either lower or higher-intensity IG	Diary and phone call or email	Inactive	*Significant effect* MVPA (min/d) ^O^ *No significant effect* Total PA, Light–intensity PA (min/d) ^O^	*Significant effect* Sedentary time (min/d) ^O^	Significant between-group increase in MVPA and decrease in sedentary time favoring the low-intensity intervention group	Low * (+,+,+,+,+)
Mendoza, 2017 [40]	Activity tracker with an app to show goal progression and text messages for PA encouragement	Facebook group and phone call	Inactive	*No significant effect* MVPA (min/d) ^O^	*No significant effect* SB (min/d) ^O^	No significant between-group changes in outcome	Low * (+,+,+,+,+)
Singh, 2020 [53]	Activity tracker for self-monitoring and manage PA maintenance following supervised exercise intervention	In-person session and printed material	Active	*Significant effect* Walking, moderate PA, MVPA, total activity (min/w) ^S^, Vigorous PA and MVPA (min/w) ^O^ *No significant effect* Vigorous PA (min/w) ^S^, Moderate PA (min/w) ^O^, Step counts (steps/d) ^O^	Not applicable	Significant between-group increase in MVPA ^S^, total activity ^S^, moderate PA ^S^, walking ^S^, MVPA ^O^ and vigorous PA ^O^ favoring the intervention group	Some concerns * (+,+,+,?,+)
Valle, 2017 [51]	Activity tracker interfaced with an app to assess PA and providing feedback on PA	In-person session, wireless weighing scale and email	Active	*No significant effect* Energy expenditure (kcal/w) ^S^	Not applicable	No significant between-group changes in outcome	Low * (+,+,+,+,+)

^S^: subjective; ^O^: objective; kcal/w: kilocalories per week; METs/w: total metabolic equivalent per week; min/d: minutes per day; min/w: minutes per week; steps/d: step counts per day; steps/w: step counts per week. * Assessed using RoB2.0 (Randomization process, deviations from intended interventions, missing outcome data, measurement of the outcome, selection of the reported result); +: low-risk of bias; /: some concerns; -: high-risk of bias. ** Assessed using ROBINS-I (bias due to confounding, bias in the selection of participants, bias in classification of interventions, bias due to deviations from intended interventions, bias due to missing data, bias in the measurement of outcomes, bias in the selection of reported results). +: low-risk of bias; /: moderate risk of bias; ?: serious risk of bias; -: critical risk of bias: NI: No information.

**Table 3 ijerph-18-05798-t003:** Physical activity (PA) and sedentary behavior (SB) interventions effects were reported in non-controlled trials.

**mHealth Only Intervention**						
**First Author, Year**	**mHealth Intervention**		**Effects**			**Risk of Bias**
			**PA**	**SB**	**Results Summary**	
Cheong, 2018 [59]	Activity tracker with an app providing exercise program		*No significant effect* Total metabolic equivalent (METs/w) ^S^	Not applicable	No significant change pre to post-intervention in an outcome	High * (?,?,?,?,?,?,+)
Kim, 2020 [60]	Activity tracker with app prescribing exercise program		*Significant effect* Total metabolic equivalent (METs/w) ^S^	Not applicable	Significant increase of METs pre to post-intervention	No information * (NI,?,+,+,?,?,+)
Lozano-Lozano, 2019 [67]	App to monitor and provide feedback on health behaviors		*Significant effect* MVPA weekday (min/d) ^O^ *No significant effect* MVPA weekend, MVPA global (min/d) ^O^ Steps weekday, steps weekend, steps global (steps/d) ^O^	Not applicable	Significant increase of MVPA weekday pre to post-intervention	No information * (NI,?,+,+,+,?,+)
Puszkiewicz, 2016 [68]	App to provide PA program		*Significant effect* Strenuous PA, mild PA (min/w)^S^ *No significant effect* Moderate PA (min/w) ^S^	Not applicable	Significant increase in strenuous PA and a significant decrease in mild PA pre to post-intervention	High * (?,?,+,+,+,?,+)
**mHealth and multicomponent intervention**						
**First Author, Year**	**mHealth intervention**	**Non-mHealth Intervention**	**Effects**			**Risk of Bias**
			**PA**	**SB**	**Results Summary**	
**mHealth without personal contact intervention**						
Le, 2017 [41]	Motivational activity tracker to record PA	Website and exercise recommendations	*No significant effect* MVPA (min/d, min/wk) ^O^ Proportion of average total daily time doing MVPA and MVPA for at least 10 min day/w ^S^	*No significant effect* Proportion of average screen time/day watching TV and days playing computer, video games, surfing the Internet ^S^	No significant change pre to post-intervention in either outcome	No information * (NI,?,?,?,?,?,+)
McCarroll, 2015 [50]	App giving motivational feedback	Weighing scale	*No significant effect* PA times (min/w) ^S^, Caloric expenditure (kcals/w) ^S^	Not applicable	No significant change pre to post-intervention in either outcome	No information * (NI,?,+,+,+,?,+)
Trinh, 2018 [62]	Activity tracker to assess PA and provide alert to decrease SB	Web-based intervention	*Significant effect* MVPA (min/w) ^S^, Step counts (steps/w) ^O^ *No significant effect* Light PA (min/w) ^O^, Number of bouts spent in MVPA ≥ 10 min	*Significant effect* Sedentary time (min/w) ^S^ *No significant effect* Total time spent in SB bouts of ≥30 min (min/w) ^O^, Number of breaks in time spent in SB bouts of ≥30 min	Significant increase of MVPA and step counts with significant reduction of sedentary time pre to post-intervention	No information * (NI,?,+,+,?,?,+)
**mHealth with personal contact intervention**						
Delrieu, 2020 [63]	Activity tracker and app to monitor activity level	In-person or phone sessions	*Significant effect* Domestic PA (min/w) ^S^ *No significant effect* Total PA, recreational PA, moderate PA, vigorous PA, walking PA (MET-min/w) ^S^, Participants proportions in low, moderate, and vigorous PA, *n* (%)	*Significant effect* Sitting time (min/w) ^S^	Significant decrease in sitting time with significantly favoring a decrease in domestic PA pre to post-intervention	High * (-,?,+,+,+,?,+)
Gell, 2017 [43]	Activity tracker for self-monitoring activity and text messages to support PA	Phone call	*No significant effect* MVPA (min/w) ^O^, Step counts (steps/d) ^O^	Not applicable	No significant change pre to post-intervention in either outcome	No information * (NI,?,+,+,+,?,+)
Ovans, 2018 [45]	Activity tracker to show feedback and progress toward a goal	In-person or phone coaching	*No significant effect* Step counts (steps/d) ^O^, Leisure score index ^S^	Not applicable	No significant change pre to post-intervention in either outcome	High * (?,?,+,+,?,?,+)
Short, 2018 [55]	Recommended app according to individual characteristics	In-person session, printed material and a phone call or email	*No significant effect* MVPA, walking, moderate PA, vigorous PA (min/w) ^S^	Not applicable	No significant change pre- to post-intervention in either outcome	No information * (NI,?,+,+,+,?,+)
Spark, 2015 [54]	Text messages for promoting behavioral change.	Phone call	*No significant effect* MVPA (min/d) ^O^	Not applicable	No significant change pre to post-intervention in an outcome	High * (?,?,+,+,+,?,+)

^S^: subjective; ^O^: objective; min/d: minutes per day; min/w: minutes per week; steps/d: step counts per day; kcals/w: kilocalories per week. * Assessed using ROBINS-I (Bias due to confounding, Bias in election of participants, Bias in classification of interventions, bias due to deviations from intended interventions, bias due to missing data, bias in measurement of outcomes, Bias in selection of reported results). +: low-risk of bias; /: moderate risk of bias; ?: serious risk of bias; -: critical risk of bias: NI: No information.

## Data Availability

The datasets used during the current study are available from the corresponding authors on reasonable request.

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
