# Peer review of "mHealth Interventions to Address Physical Activity and Sedentary Behavior in Cancer Survivors: A Systematic Review"

_ijerph, 2021, doi:10.3390/ijerph18115798_

Round 1

Reviewer 1 Report

Thank you for submitting the manuscript to the IJEHRPH. 

The subject of this review has significance in practice and is much of a necessity. This kind of work consumes lots of time, effort, and energy from the authors and this review employed many standardized tools to provide meaningful findings. 

Hope the authors find a few minor suggestions made below helpful for refining the manuscript. 

  1. Introduction: it is well-written overall, but reads a bit lengthy though.
    It would be easier for the readers to follow if this section could be shortened. 
  2. Methods: The authors described that language used was not limited to this review. There was a description regarding in which countries individual studies were conducted but no information on how many different languages were used in the publication and how those studies were interpreted. All of them were English?   
  3. Results:
    1) the main concern was what "personal contact" actually meant in this review. It is also directly associated with a question of what kind of characteristics of "personal contact" could exert positive impacts on study participants. Personalized, or "customized" features of such contacts? or "social" aspects of such interventions as the authors argued in the Discussion? In the same vein, group meetings may not have the same impacts on PA/SB as that of phone consultation does. 
    It will be more helpful for the journal readers if the authors could add some insights on the issue suggested here. 
    2) It was also wondered if the length of the intervention could make difference in the interpretation of the findings. It is acknowledged that this review is not a meta-analysis but intervention studies, especially those targeting physical activity require a certain period of time to enhance lifestyle changes.   

Reviewer 2 Report

Thank you for giving me the opportunity to review this paper. I hope that the authors find my comments productive and that they help them to improve their research work.

In this paper the authors carry out a literature review on mhealth devices and how they intervene in promoting physical exercise or reducing sedentary lifestyles in cancer survivors.

The proposed Keywords are correct and correspond to the research topic but the authors are asked to look for an alternative word to the keyword sitting and exercise (maybe replace for physical activity) and to include the keyword cancer.

In the Introduction all the keywords that will be used in the paper should be explained and the authors should indicate the main motivations for carrying out this research. Likewise, the objective of the paper should be explained in the penultimate paragraph of this section and the structure of the paper should be explained in the last paragraph.

It is necessary to include in the paper a section that makes reference to the Literature Review and where the main studies and research related to the research topic and that have given rise to the authors to carry out their research are collected. It is also necessary in this same section to include the research questions, i.e., what was the problem or gap that has been detected and about which we have made a prediction that later, in the methodology, we have been able to corroborate or not. In addition, authors are asked to bear in mind that the research questions should be based on the literature review

In Materials and Methods, the selection and criteria used to carry out the literature review are correctly explained. Likewise, the databases and keywords used for the extraction of the documents are detailed. However, the authors are asked to define the acronym RCTs.

In addition, the authors are asked to elaborate on the theoretical models on which this methodology was based, such as those carried out by the following authors Reyes-Menendez, A., Saura, J. R., & Palos-Sanchez, P. (2020). Identifying key performance indicators for marketing strategies in mobile applications: a systematic literature review. International Journal of Electronic Marketing and Retailing, 11(3), 259-277.

Authors are asked to correctly cite authors on lines 200, 201, 225, 338, 388.

The results are correctly explained and represented in a table, however authors are asked to define what MVPA is (line 268) when it first appears in the text.

In the results, the authors should check the research questions formulated (see previous comment on this) with the results obtained correctly including also a brief summary of the main conclusions obtained with the results.

Regarding Discussion, the authors justify and interpret the findings obtained with the research and support them in previous studies. Likewise, the limitations of the study are indicated and several lines for future research are presented, leaving the object of study open to future studies.

In the Conclusion section, all the information provided at the beginning of the section is irrelevant in this section. Authors are requested to include this information in Discussion. The Conclusion section is intended to clarify the objectives of the research, to state the purpose of the research and to show what the authors are demonstrating with their research. Authors are asked to develop thissection accordingly.

References are correct being mostly up to date (from 2016 to present).

Round 2

Reviewer 2 Report

The authors have adequately addressed the suggestions and the paper can be considered for publication